# Dual-view Pyramid Network for Video Frame Interpolation

Submission Id: 4636

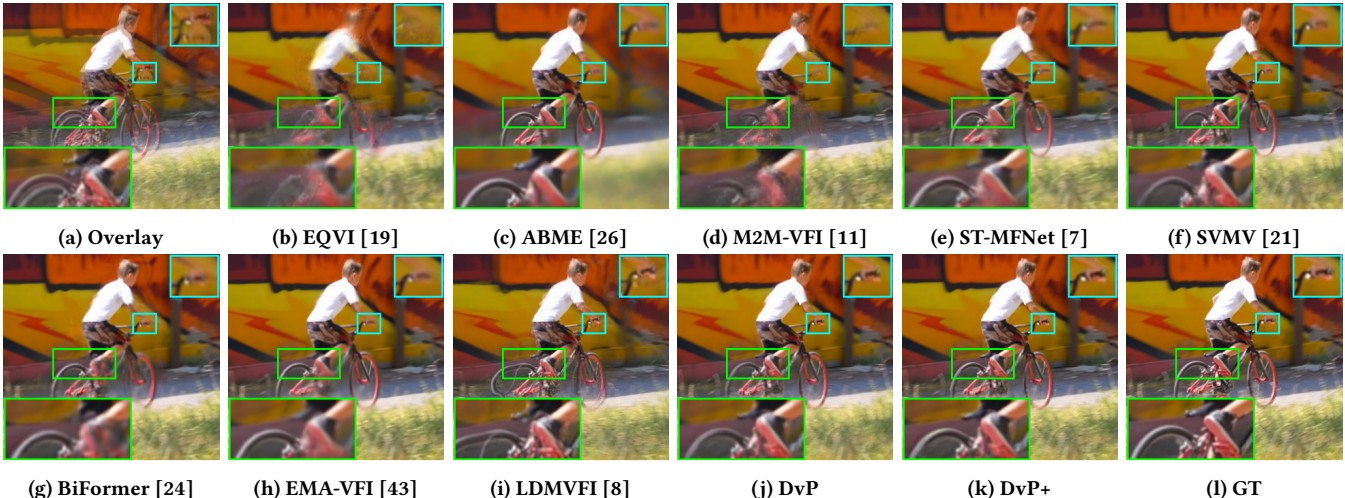

**Figure 1: An example of video frame interpolation on DAVIS [28] dataset, containing complex mixture of camera and bicycle riding motions as shown by the overlay of reference frames in (a). Previous methods yield blur or distortions at the limb of the rider and in the background as shown in (b-i), while the proposed DvP and DvP+ model synthesize clear contents (j-k).**

## ABSTRACT

Video frame interpolation is a critical component of video streaming, a vibrant research area dealing with requests of both service providers and users. However, most existing methods cannot handle changing video resolutions while improving user perceptual quality. We aim to unleash the multifaceted knowledge yielded by the hierarchical views at multiple scales in a pyramid network. Specifically, we build a dual-view pyramid network by introducing pyramidal dual-view correspondence matching. It compels each scale to actively seek knowledge in view of both the current scale and a coarser scale, conducting robust correspondence matching by considering neighboring scales. Meanwhile, an auxiliary multiscale collaborative supervision is devised to enforce the exchange of knowledge among current scale and a finer scale and thus reduce error propagation from coarse to fine scales. Based on the robust video dynamic caption of pyramidal dual-view correspondence matching, we further develop a pyramidal refinement module that formulates frame refinement as progressive latent representation generations by developing flow-guided cross-scale attention for feature fusion among neighboring frames. The proposed method achieves favorable performance on several benchmarks of varying video resolutions with better user perceptual quality and a relatively compact model size.

## CCS CONCEPTS

• **Computing methodologies → Image and video acquisition**.

## KEYWORDS

Video Frame Interpolation, Dual-view Pyramid, Collaborative Supervision

## 1 INTRODUCTION

Video frame interpolation analyzes video dynamics to synthesize intermediate frames between reference frames [1]. As a critical video processing technique, it has various applications in multimedia systems, such as video compression [14, 40], video on demand streaming [29] and video editing [3, 4, 15, 34]. Recent years have witnessed emergence of approaches leveraging deep learning algorithms for video frame interpolation [8, 12, 43]. However, video frame interpolation is still challenged by video dynamics that involve complex occlusions and large motions [11, 12].

One way deployed by existing video frame interpolation methods to understand video dynamics is exploring motion cues like optical flow among reference frames. Based on optical flow estimates between reference frames and the intermediate frame to be synthesized, reference frames at two time steps are sampled via warping and fused to generate interpolation results at a time step in-between [1]. Advances in optical flow estimation [2, 17, 36, 38] have been leveraged to handle complex video dynamics in recent video

frame interpolation methods [11, 19]. To mitigate degradation of interpolation quality due to deficient quality of optical flow estimation, asymmetric [26] and multiple flow fields estimations [7, 11, 21] has been considered. But video frame interpolation for varying resolution videos remains difficult, where larger motions need to be handled with longer-range correlation analysis while avoiding error propagation in this process [24].

To tackle these difficulties, we propose to capture long-range correlations among reference frames and intermediate frames to be synthesized with pyramidal dual-view correspondence matching. Our goal is to build a pyramid network that is capable of handling complex occlusions and motions while preserving this capability for varying video resolutions through unleashing the multifaceted knowledge yielded by the hierarchical views at multiple scales. We approach this by introducing dual-view correspondence matching in a pyramidal network architecture that estimates multiple bidirectional flow fields and their corresponding weights from coarse to fine, where each scale considers neighboring scales for correspondence matching. This enables knowledge in view of both the current scale and a coarser scale coalesce into more robust correspondence matching for analyzing long-range correlation. Unlike some other multimedia tasks such as optical flow estimation [45] and video object segmentation [10], the correspondence matching in video frame interpolation is subject to the intermediate frame which is not available for the computation of correspondence matrix. As a result, calibration of correspondence matching for this unknown frame plays a critical role. Therefore the proposed pyramidal architecture allows the dual-view correspondence matching to be gradually calibrated according to the intermediate frame to be synthesized through estimates of multiple bidirectional flow fields and their corresponding weights across scales. As the calibration is operated in RGB color space through the generation of intermediate frames by directly sampling reference frames with estimates of multiple bidirectional flow fields and their corresponding weights across scales, the quality of supervision offered by downscaled ground truth intermediate frames at coarser scales may not be sufficient to calibrate the dual-view correspondence matching. For this reason, we introduce an auxiliary multi-scale collaborative supervision to enforce the exchange of knowledge among current scale and a finer scale, thus reducing error propagation across scales.

Compared with existing attention-based video frame interpolation methods [24, 43] that incorporate long-range correlation by leveraging the capability of Transformer blocks for long-range connectivity, we pay attention to the multifaceted knowledge yielded by the hierarchical views of a pyramidal network architecture. Establishing robust dense correspondences among frames via the proposed pyramidal dual-view correspondence matching instead of Transformer blocks exhibits three potential advantages. Firstly, pyramidal network architecture disseminates the burden of establishing long-range connectivity across scales, thus it facilitates the design of a compact model, gradually building global connectivity across scales. Secondly, pyramidal dual-view correspondence matching can utilize a broad range of supervisions from multi-scale knowledge which are critical for model generalization among varying video resolutions. Thirdly, we can enhance representation capability of the proposed pyramidal network architecture by further incorporating latent representations of reference frames

and intermediate frames to be synthesized. This is achieved by developing a refinement module, which only considers a relatively small neighborhood to efficiently refine the synthesis of each pixel utilizing the guidance of video dynamics captured by the pyramidal dual-view correspondence matching. To take advantage of the robust video dynamics captured by pyramidal dual-view correspondence matching in constructing latent representations, we introduce a pyramidal refinement module that formulates frame refinement as progressive latent representation generations. Latent representations of reference frames are adaptively transformed to establish the latent representation of the intermediate frame to be synthesized by our proposed flow-guided cross-scale attention for feature fusion among neighboring frames. We show the proposed Dual-view Pyramid (DvP) network and its enhanced version DvP+ with our pyramidal refinement module are able to handle changing video resolutions while improving user perceptual quality. The main contributions of this paper are summarized as follows:

- We introduce pyramidal dual-view correspondence matching for video frame interpolation, which enables efficient capture of long-range correlations to handle complex video dynamics along with large motion by seeking knowledge in view of both the current scale and a coarser scale.
- We propose an auxiliary multi-scale collaborative supervision for effective learning of pyramidal dual-view correspondence matching, which reduces error propagation across scales to improve video frame interpolation quality for varying video resolutions.
- We combine designs above in building Dual-view Pyramid (DvP) network for video frame interpolation, and introduce a pyramidal refinement module to build DvP+ that enhances representation capacity using latent representation generations through flow-guided cross-scale attention.
- We demonstrate that DvP network and its enhanced version achieve favorable video frame interpolation performance on several benchmarks with video resolutions ranging from 480p to 4K, while keeping a relatively compact model size.

## 2 RELATED WORKS

### 2.1 Video frame interpolation

Video frame interpolation is a classical task in video processing [1, 42] that attracts increasing attention due to the emergence of streaming services and a growing demand for better video systems. Existing methods leverage deep neural networks to improve the performance of video frame interpolation on several benchmarks [7, 8, 11, 12]. The majority of these approaches belong to the flow-based paradigm [19, 22, 32] that estimates flow fields among frames to synthesize intermediate frames and thus rely on high quality flow estimates to mitigate corruption in output, and kernel-based paradigm [18, 23] that predicts adaptive convolution kernels to generate intermediate frames and may be vulnerable to blurry output by lacking explicit flow guidance. Recently, progress has been made by improving quality of flow estimation and compensate imperfect estimates. This includes generating asymmetric or multiple flow estimates [7, 11, 21, 26], deploying sophisticated enhancement, leveraging Transformer-based architecture [24, 43] and latent diffusion model [8]. For example, ST-MFNet [7] introduces

compensation with a multi-branch structure and a 3D convolution-based enhancement network. However, these methods are still deficient to handle large motion in complex video dynamics especially regarding improvement of perceptual quality [8], and are limited by computational burden due to deployment of heavy enhancement module, Transformer or latent diffusion. As a result, we propose to explore effective analysis of video dynamics via more robust correspondence matching approach, unleash the multifaceted knowledge yielded by multiple scales in a pyramid network.

## 2.2 Correspondence matching

Establishing spatiotemporal correspondences among frames has been widely studied for various multimedia applications. For example, correspondence matching is involved in binocular stereo to determine the disparity of pixels in stereo pair [30], and in fine-grained action recognition to facilitate the capture of contextual details from multiple perspectives [37]. This technique is also applied in optical flow estimation [45] and video object segmentation [10] to estimate the motion of scene contents across frames. Acquiring high quality correspondence matching is nontrivial, due to the ill-posedness of the task. To this end, Patchmatch Stereo++ introduces deep-learning-based continuous disparity optimization to tackle edge ambiguity [30], MVFlow extracts pre-computed information in video compression as a robust prior to mitigate ambiguities in optical flow estimation [45]. The ambiguities for correspondence matching in video frame interpolation is distinguished by the absence of intermediate frame to compute correspondences. BMBC [25] and XVFI [32] devised respectively bilateral cost volume and complementary flow reversal for correspondence matching in video frame interpolation. These approaches are limited by linear motion assumption, motivating ABME [26] and NCM [14] being developed to handle asymmetric motions and inaccurate flows. However, video frame interpolation for varying resolution videos still challenges correspondence matching due to larger motions that requires longer-range correlation analysis while avoiding error propagation in this process. In this paper, we disseminate the burden of establishing long-range connectivity across scales by introducing pyramidal dual-view correspondence matching, mitigating error propagation through knowledge exchange while gradually building global connectivity across scales.

## 3 METHODS

Figure 2 shows an overview of the proposed approach. The pyramidal video frame interpolation network composed of two subnetworks. Given four consecutive reference frames $\{I_{-1}, I_0, I_1, I_2\}$, the first subnetwork $H_1$ establishes robust dense correspondences among reference frames and intermediate frame $I_t, t \in (0, 1)$ to be synthesized between $I_0$ and $I_1$ with pyramidal dual-view correspondence matching and an auxiliary multi-scale collaborative supervision, and the second subnetwork $H_2$ progressively constructs latent representations to refine the synthesis of each pixel of $I_t$ through a pyramidal refinement module with the guidance of video dynamics generated by $H_1$.

## 3.1 Pyramidal network with dual-view correspondence matching

To capture complex video dynamics among $\{I_{-1}, I_0, I_1, I_2\}$ and $I_t, t \in (0, 1)$ for video frame interpolation, the first subnetwork is developed to estimate $M$ candidate pairs of flow fields $\{u_{0 \to t}^{k,m}, u_{1 \to t}^{k,m}\}_{m=1}^{M}$ and their corresponding weights $\{w_0^{k,m}, w_1^{k,m}\}$, which yield estimate of intermediate flow fields $u_{t \to 0}^{k}, u_{t \to 1}^{k}$ and $I_t^{k}$ for each level $k$ via direct sampling. Let $\boldsymbol{p}_t$ and $\boldsymbol{q}_0$ denote pixel coordinates, the intermediate flow estimate $u_{t \to 0}^{k}[\boldsymbol{p}_t]$ is computed through weighted sampling of $\{u_{0 \to t}^{k,m}\}_{m=1}^{M}$ as follows:

$$\frac{\sum_{m=1}^{M} \sum_{\forall \boldsymbol{q}_0 \in I_0^k} w_0^{k,m}[\boldsymbol{q}_0] B(\boldsymbol{q}_0 + u_{0 \to t}^{k,m}[\boldsymbol{q}_0] - \boldsymbol{p}_t)(-u_{0 \to t}^{k,m}[\boldsymbol{q}_0])}{\sum_{m=1}^{M} \sum_{\forall \boldsymbol{q}_0 \in I_0^k} w_0^{k,m}[\boldsymbol{q}_0] B(\boldsymbol{q}_0 + u_{0 \to t}^{k,m}[\boldsymbol{q}_0] - \boldsymbol{p}_t)},$$

(1)

where $B(\cdot)$ denotes the bilinear interpolation kernel, and $u_{0 \to t}^{k,m}$ is parameterized with estimates of flow fields $u_{0 \to -1}^{k,m}, u_{0 \to 1}^{k,m}$ using the quadratic motion model [41]to accommodate nonlinear motion as:

$$u_{0 \to t}^{k,m} = (u_{0 \to 1}^{k,m} + u_{0 \to -1}^{k,m})t^2/2 + (u_{0 \to 1}^{k,m} - u_{0 \to -1}^{k,m})t/2.$$

(2)

By analogy, we compute the estimate of $u_{t \to 1}^{k}[\boldsymbol{p}_t]$ via weighted sampling of $\{u_{1 \to t}^{k,m}\}_{m=1}^{M}$, and the estimate of intermediate frame $\tilde{I}_t^{k}[\boldsymbol{p}_t]$ via weighted sampling of $\{I_0^k, I_1^k\}$. Such weighted sampling warps and fuses pixels direcly in the RGB color space, thus the estimation results are directly affected by the quality of estimates of candidate pairs of flow fields and their corresponding weights. However, acquiring estimates of sufficient quality is challenged by video dynamics that involve complex and large motions [11, 12]. To tackle this challenge, we propose pyramidal network with dual-view correspondence matching to improve video frame interpolation quality of complex video dynamics with varying video resolutions.

Based on the observation that large motion is easier to be captured at coarser scales while small object may only be observed at finer scales in a pyramidal network, the core idea of pyramidal dual-view correspondence matching is to simultaneously acquire knowledge in view of both the current scale and a coarser scale from coarse to fine. In detail, let $H_1$ utilizes a $K$-level feature encoder $f$ [13] to extract pyramidal features $\{F_i^k\}_{k=1}^K, i \in \{0, -1, 1\}$ of reference frames, to enable multifaceted knowledge yielded by the hierarchical views at multiple scales to coalesce into more robust correspondence matching for analyzing long-range correlation of complex video dynamics, we compute the correspondences with reference frames $I_i, I_j$ for a pixel through search grids at current scale and a coarser scale $c_{i \to j}^k(\boldsymbol{q}_i, \boldsymbol{\delta}^k, \boldsymbol{\delta}^{k+1})$ as follows:

$$(F_i^k[\boldsymbol{q}_i] \cdot F_j^k[B(\boldsymbol{q}_i + \sum_{m=1}^{M} w_i^{k,m}[\boldsymbol{q}_i] u_{i \to j}^{k,m}[\boldsymbol{q}_i] + \boldsymbol{\delta}^k)]) \odot$$

(3)

$$(F_i^k[\boldsymbol{q}_i] \cdot S_{k+1}(F_j^k)[B(\boldsymbol{q}_i/2 + \sum_{m=1}^{M} w_i^{k,m}[\boldsymbol{q}_i] u_{i \to j}^{k,m}[\boldsymbol{q}_i]/2 + \boldsymbol{\delta}^{k+1})])$$

where $\cdot$ and $\odot$ refer to vector dot product and channel concatenation respectively, $\boldsymbol{\delta}^k, \boldsymbol{\delta}^{k+1}$ denotes the displacements in the correlation search grid of size $D^2$ at scale $k$ and a coaser scale $k + 1$. $S_{k+1}(\cdot)$

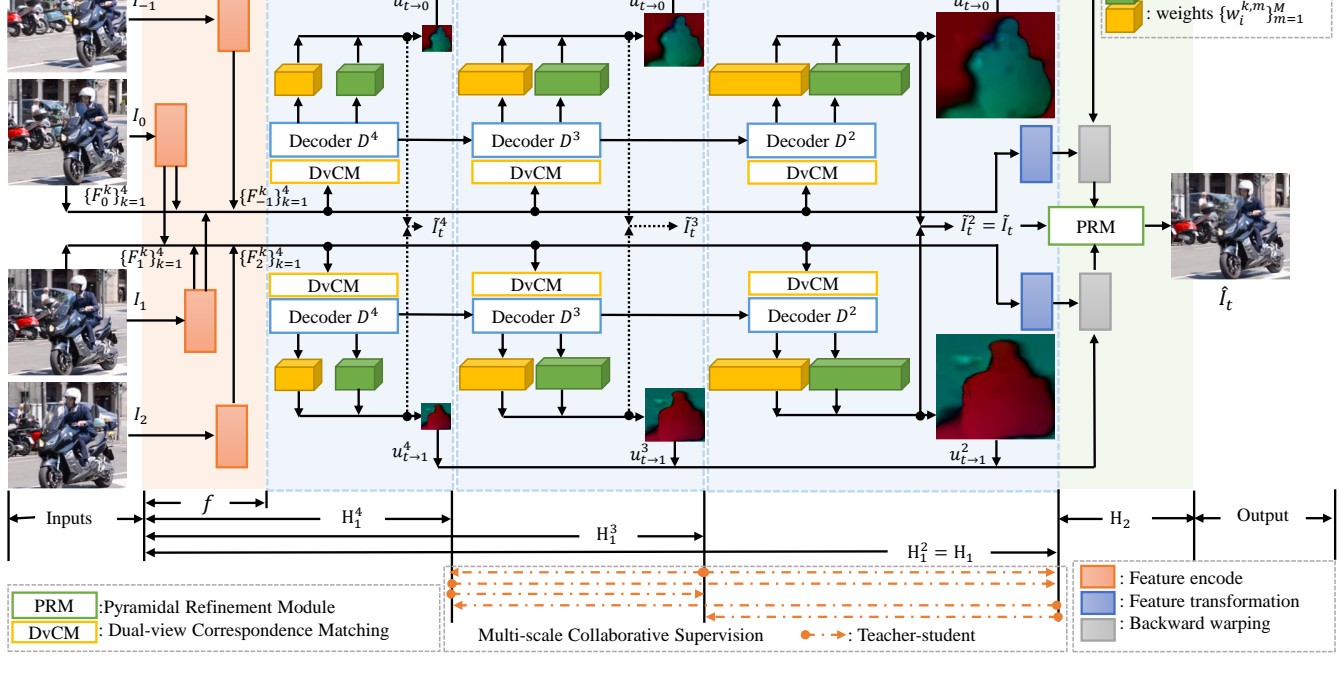

**Figure 2: Overview of the Dual-view Pyramid (DvP) network with the auxiliary multi-scale collaborative supervision for video frame interpolation. We give a concise demonstration using a 4-level multi-scale network. Each reference frame $I_i$ is transformed into pyramidal features $\{F_i^k\}_{k=1}^4$. Based on dual-view correspondence matching, the decoder at each level yields estimates of multiple bidirectional flow fields and their corresponding weights which are utilized to compute intermediate flow $u_{t\to i}^k$ and direct sampling results $\tilde{I}_t^k$. $H_2$ synthesizes the intermediate frame $\hat{I}_t$ through a pyramidal refinement module. The dashed paths denotes knowledge exchange among estimators $\{H_1^k\}_{k=2}^4$ across scales.**

conduct bilinear resizing to yield features with the same spatial dimension of $F^{k+1}$.

Different from correspondence analysis through residual-based single connectivity [21] or Transformer-based dense connectivity [24, 43], the proposed matching allows knowledge of correspondences with different ranges being accessible and adaptable across scales via dual-view-based sparse connectivity, so a compact model is capable to handle complex scene dynamics to meet the need of video frame interpolation for varying video resolutions.

## 3.2 Multi-scale collaborative supervision

Noting that the correspondence matching in video frame interpolation is subject to the intermediate frame, which is different from correspondence matching subject to only reference frames [36]. We propose to gradually calibrate the dual-view correspondence matching across scales according to the intermediate frame to be synthesized during training by generating its estimate with direct sampling based on multiple bidirectional flow fields and their corresponding weights from coarse to fine.

As the calibration is operated in RGB color space through the generation of intermediate frames by directly sampling, we develop auxiliary multi-scale collaborative learning across scales to improve the quality of supervision at coarser scales for better calibration

of the proposed pyramidal dual-view correspondence matching by exchanging knowledge, and thus reducing error propagation across scales.

To formulate the loss function, denote intermediate motion estimations between the intermediate frame $I_t$ to be synthesized and the input reference frames $\{I_0, I_1\}$ for level $k$ as:

$$u_{t\to 0}^k = H_1^k(I_0, I_{-1}, I_1, t) = H_1^k(x_{0\to t}), \quad (4)$$

$$u_{t\to 1}^k = H_1^k(I_1, I_2, I_0, 1-t) = H_1^k(x_{1\to t}), \quad (5)$$

where a smaller $k$ represents a finer level. Considering subnetwork $H_1$ being composed of $K-1$ nested estimators $\{H_1^k\}_{k=2}^K$ with the finest-scale estimator being $H_1^2 = H_1$, we conduct estimation of $\{u_{t\to 0}^k\}_{k=2}^K$ with the proposed pyramidal dual-view correspondence matching to establish robust correspondence among frames, using feature pyramids generated in a similar way as LiteFlowNet [13]. We define the intermediate flow estimator for level $k$ as $H_1^k = D^k \circ D^{k+1} \circ \cdots \circ D^K \circ f$ where $D^k$ is composed of a motion decoder $D_u^k$ and a weight decoder $D_w^k$. Leveraging robust correspondence among frames, the motion decoder generates $M$ candidate flow fields $\{u_{0\to t}^{k,m}\}_{m=1}^M$, and the weight decoder provides combination weight $w^{k,m}$ for each candidate. Both decoders are designed to

generate upsampled estimates, i.e., estimates with the same spatial resolution of level $k-1$ for an image pyramid $\{I_i^k\}_{k=1}^K$.

Given $N$ samples with the sample index $n$, we refer to respectively $x_{0 \to t,n}$ and $x_{1 \to t,n}$ as regular inputs for the learning of the intermediate flow estimations $u_{t \to 0}^k$ and $u_{t \to 1}^k$, and $x_n^* = I_{t,n}$ as the privileged information. The question is that how we exploit the privileged information to build better flow estimators $\{H_1^k\}_{k=2}^K$ for test time. To this end, the proposed auxiliary multi-scale collaborative supervision jointly learns teacher $H_1^T$ using the regular input-privileged information pairs, computes the teacher output $y_{t \to i,n}^T = u_{t \to i,n}^T, i \in \{0,1\}$, and learns student $H_1^k$ using both the regular input-privileged information pairs and the regular input-teacher output pairs with imitation parameter $\lambda_{IM} \in [0,1]$. Accordingly, the loss of auxiliary multi-scale collaborative supervision for sample $n$ is computed as:

$$L_n^{auxiliary} = \sum_{T=2}^K \sum_{k=2}^K \lambda_{IM} L_n^{IM,T,k} + (1-\lambda_{IM}) L_n^{non-IM,k} \quad (6)$$

$$= \sum_{k=2}^K [(1-\lambda_{IM})(K-1) L_n^{non-IM,k} + \lambda_{IM} \sum_{T=2}^K L_n^{IM,T,k}],$$

$$L_n^{non-IM,k} = \lambda_{PR} L_n^{PR,k}(u_{t \to 0,n}^k, u_{t \to 1,n}^k) + \rho(\tilde{I}_{t,n}^k, I_{t,n}^k), \quad (7)$$

where $L_n^{IM,T,k}$ is the imitation loss for the errors of student $H_1^k$ with respect to teacher $H_1^T$. $L_n^{non-IM,k}$ is the non-imitation loss for the errors of $H_1^k$, including the per-pixel minimum photometric reprojection loss $L_n^{PR,k}$ with parameter $\lambda_{PR}$ for the errors of backward warped image pyramids of input frames $\{I_{t \to 0,n}^k, I_{t \to 1,n}^k\}$ according to the intermediate flow estimates with respect to the true intermediate frame, and the errors of the direct sampling results with respect to the ground truth intermediate frame. $\rho$ denotes the Charbonnier penalty.

We formulate the imitation loss by considering the error distribution of teachers' predictions. A teacher's prediction is adaptively chosen according to its reliability that is evaluated using the empirical photometric reprojection error. We define the reliability $r_{t \to 0,n}^k$ of intermediate flow estimate $u_{t \to 0,n}^k$ to be 1 whenever the error of $I_{t \to 0,n}^k$ with respect to $I_{t,n}^k$ is smaller than that of $I_{t \to 1,n}^k$, and 0 otherwise. The reliability $r_{t \to 1,n}^k$ of intermediate flow estimate $u_{t \to 1,n}^k$ is defined analogously to $r_{t \to 0,n}^k$. The imitation loss is computed as:

$$L_n^{IM,T,k} = \sum_{i=0}^1 S_k(r_{t \to i,n}^T) \rho(H_1^k(x_{i \to t,n}), S_k(y_{t \to i,n}^T)), \quad (8)$$

where bilinear resizing $S_k(\cdot)$ is used to ensure that the spatial dimension of teacher is the same as that of the student $H_1^k$. Let $L_n^{IN}$ denote the interpolation loss of the second subnetwork, the total loss for the first subnetwork that composed of a pyramidal network with dual-view correspondence matching is formulated as follows.

$$L = \frac{1}{N} \sum_{n=1}^N (L_n^{IN} + L_n^{auxiliary}), \quad (9)$$

$$L_n^{IN} = \rho(\tilde{I}_{t,n}^k, I_{t,n}^k) + \lambda_{VGG} \|\phi(\tilde{I}_{t,n}^k), \phi(I_{t,n}^k)\|_1, \quad (10)$$

where we use features of frames $\phi(\cdot)$ extracted from $conv4\_3$ of a pretrained VGG16 [33] to construct the perceptual loss with parameter $\lambda_{VGG}$.

## 3.3 Pyramidal refinement

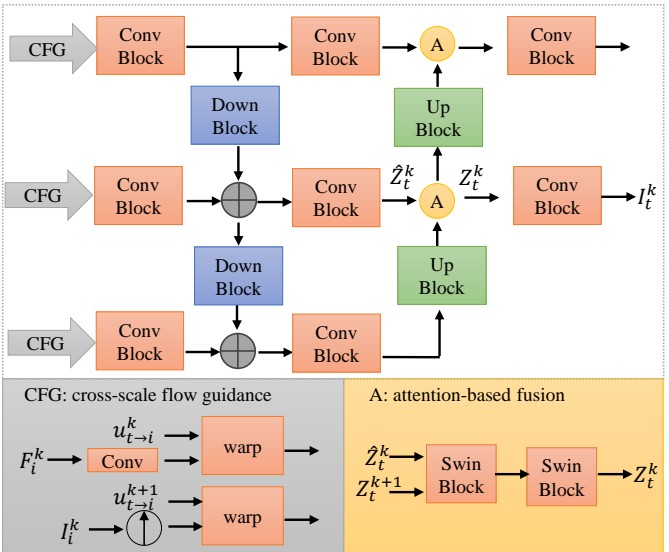

**Figure 3: Overview of the pyramidal refinement module with flow-guided cross-scale attention.**

Since $H_1$ estimates intermediate frame via direct sampling in color space, its capability of representing complex dynamics may be deficient due to low quality reference frames and flow estimates, causing degradation of video frame interpolation quality. To address this limitation, we develop subnetwork $H_2$ that progressively constructs latent representations to refine the estimate of $I_t$ with direct sampling. $H_2$ is formed by a pyramidal refinement module that deploys the guidance of video dynamics yielded by $H_1$. Instead of using a convolution-based pyramidal network with each scale operating under guidance of the intermediate flow estimation at current scale [26], we propose to improve the capability of capturing long-range correlations among reference frames and intermediate frame estimates in latent space with flow-guided cross-scale attention. Specifically, to efficiently extract spatiotemporal dynamics of scene contents among reference frames with a relatively small neighborhood, each scale of the refinement module takes as input both warped information guided by intermediate flow estimates of current scale and a coarser scale. As shown in Figure 3, the extracted scene contents are adaptively transformed according to intermediate flow guidance, based on which we formulate frame refinement as progressive latent representation generations from coarse to fine where attention-based feature fusion among neighboring frames is conducted with Swin transformer blocks [20].

We deploy an interpolation loss similar to equation (10) to supervise the second subnetwork, enforcing smaller difference between estimates of intermediate frame refined by the pyramidal refinement module $\hat{I}_t$ instead of $\tilde{I}_t$ and the ground truth intermediate frame to be synthesized.

# 4 EXPERIMENTS

## 4.1 Experimental Setup

The proposed method is implemented with Pytorch [27], using publicly available datasets for training and testing. Our models are trained on samples generated using a mixture of $64, 612$ sequences of 7 frames from the Vimeo90K-Septuplet training dataset [42] and $15, 029$ sequences of 25 frames from the Adobe240 dataset [35, 41]. For a sequence from the Vimeo90K-Septuplet training dataset, we learn to interpolate the $4^{th}$ frame using the $1^{st}$, $3^{rd}$, $5^{th}$ and $7^{th}$ frames. For a sequence from the Adobe240 dataset, we take the $1^{st}$, $9^{th}$, $17^{th}$ and $25^{th}$ frames of each sequence as inputs and one among the $10^{th}$ to $16^{th}$ frames randomly as target to synthesize. We crop $448 \times 256$ patches and conduct image flipping and sequence reversal randomly for data augmentation. The training utilizes the Adam optimizer [16] and a batch size of 4, starting with a learning rate of $10^{-4}$ which is reduced by half once for every $2 \times 10^5$ iterations, until convergence with around $6 \times 10^5$ iterations. We use PSNR, SSIM [39] and LPIPS [44] as metrics to quantitively evaluate the performance and additionally deploy tOF [6] as metric to quantitively evaluate temporal consistency for multiple frame interpolation, following [22, 32]. Higher PSNR and SSIM values along with lower LPIPS and tOF values indicate better results.

## 4.2 Benchmarks

We evaluate the performance for single frame interpolation on $2, 849$ sequences of 7 frames with a spatial resolution of 480p from the DAVIS dataset [28] as [7, 41], $1, 164$ sequences with spatial resolution ranging from $640 \times 368$ to $1280 \times 720$ from the SNU-FILM dataset that are divided into 4 settings, *Easy*, *Medium*, *Hard* and *Extreme*, with varying difficulties defined by the temporal gap between input frames as [5], 8 sequences of 100 frames with a spatial resolution of 4K from the Xiph-4K dataset [22] as [24, 43]. To evaluate the performance for multiple frame interpolation, we utilize the X-TEST benchmark that consists of 15 scenes of 4K video resolution to interpolate 7 intermediate frames in-between reference frames as [32].

## 4.3 Comparisons with State-of-the-art

For single frame video interpolation, we compare the proposed approach against the state-of-the-art methods, including EQVI [19], ABME [26], M2M-VFI [11], ST-MFNet [7], SVMV [21], BiFormer [24], EMA-VFI [43] and LDMVFI [8]. For multiple frame interpolation, we additionally included XVFI [32] for comparisons. Our enhanced version DvP+ is different from DvP by deploying the proposed pyramidal refinement module. Tables 2 and 3 show quantitative results for single frame interpolation of varying video resolutions on the SNU-FILM [5], DAVIS [28] and Xiph-4K [22] dataset. Our approach achieves the best perceptual quality in terms of LPIPS [44] on all datasets, while achieving competitive PSNR and SSIM with a relatively compact model size. Compared with previous state-of-the-art scheme ST-MFNet [7] that achieves the best PSNR and SSIM score on most datasets, our DvP model reduces the average LPIPS by 34.03% with 68.57% less number of parameters, our DvP+ model reduces the average LPIPS by 44.50% with 60.95% less number of parameters. Especially, our DvP and DvP+ models obtain

**Table 1: Quantitative comparisons of video frame interpolation results on the X-Test [32] dataset. The best and the second-best results are marked in bold and with an underline, respectively.**

| Methods | X-Test PSNR↑ / SSIM↑ / LPIPS↓ / tOF↓ |
|---|---|
| EQVI [19] | 28.15 / 0.8086 / 0.175 / 3.58 |
| XVFI [32] | 30.12 / 0.8344 / 0.089 / 2.15 |
| ABME [26] | 30.16 / 0.8479 / 0.147 / 2.54 |
| M2M-VFI [11] | 30.06 / 0.842 / 0.086 / 1.53 |
| SVMV [21] | 30.65 / 0.8549 / 0.131 / 2.78 |
| BiFormer [24] | 31.32 / 0.8618 / 0.121 / 1.90 |
| EMA-VFI [43] | 31.45 / 0.8574 / 0.163 / 3.05 |
| LDMVFI [8] | 23.34 / 0.6753 / 0.211 / 6.52 |
| DvP | 33.27 / 0.8972 / 0.068 / 1.48 |
| DvP+ | **33.35** / **0.8982** / **0.062** / **1.21** |

respectively 54.92% and 65.98% higher LPIPS compared with ST-MFNet [7] on the challenging Xiph-4K [22] dataset that composed of 4K videos, showing the capability of the proposed approach to more effectively capturing complex and large motions in high-resolution videos. This is likely due to the proposed approach is capable of improving the quality of multiple intermediate flow estimation by leveraging the multifaceted knowledge yielded by the hierarchical views at multiple scales in a pyramid network. Compared with recent perceptually-oriented video frame interpolation scheme LDMVFI [8] that explores the power of latent diffusion model [9, 31] in synthesizing perceptually-optimized frames, our DvP+ model achieves average LPIPS reduction by 21.87% with number of parameters over 50 times less than that of LDMVFI [8], and our DvP model achieves average LPIPS reduction by 5.86% with number of parameters over 65 times less than that of LDMVFI [8]. This demonstrates that the proposed pyramidal refinement module is able to efficiently deploy the representation capability of latent space to facilitate frame synthesis in color space. The results of quantitative comparisons are consists with the visual perceptual comparisons in Figure 1 that contains complex mixture of camera and object motions, and in Figures 4 that involves moving objects with finer details. Our approach generates more visually pleasing results with less artifacts and blurry details.

Tables 1 shows quantitative results for multiple frame interpolation of 4K video resolution on the X-Test [32] dataset. Our approach achieves the best perceptual quality in terms of LPIPS [44] and best temporal consistency in terms of tOF [6] on all datasets, while achieving the best PSNR and SSIM. Compared with previous state-of-the-art scheme EMA-VFI [43] and BiFormer [24] that achieve respectively the best PSNR and SSIM score, our DvP model reduces the average LPIPS by 51.04% and average tOF by 36.80 with 1.89 dB PSNR gain on average, our DvP+ model reduces the average LPIPS by 55.36% average tOF by 48.33 with 1.97 dB PSNR gain on average. The results demonstrate that the proposed approach can be better extended to multiple frame interpolation of high video resolution for favorable performance, illustrating the proposed scheme for

**Table 2: Quantitative comparisons of video frame interpolation results on the SNU-FILM [5] datasets. The best and the second-best results are marked in bold and with an underline, respectively.**

| Methods | FILM(Easy) | FILM(Medium) | FILM(Hard) | FILM(Extreme) |
|---|---|---|---|---|
| | PSNR↑ / SSIM↑ / LPIPS↓ | PSNR↑ / SSIM↑ / LPIPS↓ | PSNR↑ / SSIM↑ / LPIPS↓ | PSNR↑ / SSIM↑ / LPIPS↓ |
| EQVI [19] | 38.75 / 0.9804 / 0.030 | 35.48 / 0.9667 / 0.050 | 30.65 / 0.9143 / 0.108 | 25.64 / 0.7968 / 0.197 |
| ABME [26] | 39.59 / 0.9827 / 0.022 | 35.77 / 0.9650 / 0.037 | 30.58 / 0.9001 / 0.066 | 25.11 / 0.7809 / 0.131 |
| M2M-VFI [11] | 39.61 / 0.9828 / 0.021 | 35.72 / 0.9651 / 0.035 | 30.30 / 0.8985 / 0.063 | 24.79 / 0.7746 / 0.127 |
| ST-MFNet [7] | **40.78** / **0.9850** / 0.019 | 37.11 / 0.9733 / 0.036 | 31.70 / 0.9213 / 0.073 | 25.81 / 0.8019 / 0.148 |
| SVMV [21] | 40.26 / 0.9836 / 0.017 | 37.14 / **0.9738** / 0.027 | 31.76 / 0.9244 / 0.059 | 25.76 / **0.8036** / 0.126 |
| BiFormer [24] | 36.58 / 0.9694 / 0.053 | 33.82 / 0.9500 / 0.069 | 29.71 / 0.8877 / 0.098 | 24.89 / 0.7722 / 0.151 |
| EMA-VFI [43] | 39.71 / 0.9834 / 0.019 | 35.95 / 0.9666 / 0.033 | 30.92 / 0.9043 / 0.060 | 25.40 / 0.7834 / 0.119 |
| LDMVFI [8] | 38.67 / 0.9784 / 0.014 | 34.00 / 0.9496 / 0.028 | 28.55 / 0.8665 / 0.060 | 23.71 / 0.7378 / 0.128 |
| DvP | 40.28 / 0.9835 / 0.014 | 37.21 / **0.9738** / 0.023 | 31.96 / 0.9243 / 0.051 | **25.85** / 0.8006 / 0.111 |
| DvP+ | 40.26 / 0.9834 / **0.012** | **37.22** / 0.9737 / **0.020** | **32.00** / **0.9250** / **0.041** | 25.79 / 0.7943 / **0.097** |

**Table 3: Quantitative comparisons of video frame interpolation results on the DAVIS [28] and Xiph-4K [22] datasets. The best and the second-best results are marked in bold and with an underline, respectively.**

| Methods | DAVIS | Xiph-4K | Params(M) |
|---|---|---|---|
| | PSNR↑ / SSIM↑ / LPIPS↓ | PSNR↑ / SSIM↑ / LPIPS↓ | |
| EQVI [19] | 27.64 / 0.8317 / 0.166 | 33.54 / 0.9003 / 0.221 | 25.4 |
| ABME [26] | 26.98 / 0.8052 / 0.145 | 33.96 / 0.9011 / 0.233 | 18.1 |
| M2M-VFI [11] | 27.28 / 0.8140 / 0.104 | 33.92 / 0.8991 / 0.212 | 7.6 |
| ST-MFNet [7] | **28.36** / **0.8438** / 0.123 | **34.93** / **0.9104** / 0.244 | 21.0 |
| SVMV [21] | 28.17 / 0.8411 / 0.099 | 34.58 / 0.9030 / 0.170 | **4.8** |
| BiFormer [24] | 26.36 / 0.7946 / 0.172 | 33.49 / 0.8952 / 0.212 | 11.2 |
| EMA-VFI [43] | 27.92 / 0.8264 / 0.099 | 34.67 / 0.9071 / 0.230 | 65.7 |
| LDMVFI [8] | 25.64 / 0.7594 / 0.103 | 31.39 / 0.8543 / 0.085 | 439.0 |
| DvP | 28.26 / 0.8406 / 0.084 | 34.41 / 0.8944 / 0.110 | 6.6 |
| DvP+ | 28.25 / 0.8394 / **0.072** | 34.26 / 0.8889 / **0.083** | 8.2 |

knowledge exchange among scales to succeed in the caption of spatiotemporal video dynamics.

## 4.4 Ablation Study

To understand the effectiveness of the proposed pyramidal dual-view correspondence matching, the auxiliary multi-scale collaborative supervision and the pyramidal refinement module, we perform an ablation study by gradually adding these components on two baselines. The results are summarized in Table 4, where the proposed model without and with a refinement module are compared with corresponding variants in the top 5 rows and bottom 3 rows respectively. The baseline without a refinement module, as shown in line 1, computes the correspondences between neighboring frames for each scale in view of current scale, using correlations among image features at that scale. The baseline with a refinement module, as shown in line 6, constructs each scale of a pyramidal module by taking warped inputs guided by flow estimates of current scale.

**Effectiveness of pyramidal dual-view correspondence matching.** For evaluation, the proposed dual-view correspondence matching is gradually involved from the coarsest level 6 to finer levels

in a pyramidal network architecture, yielding results of 3 variants as shown in line 2 to 4 of Table 4. Compared with the baseline in line 1 of Table 4, the three variants deployed the proposed dual-view correspondence matching to seek knowledge in view of both the current scale and a coarser scale instead of in view of only current scale. As such, during decoding of video dynamics, there is correspondence information among neighboring frames from both current and coaser levels for each level. At the expense of a slight increase in the numbers of model parameters, Line 1 to 4 of Table 4 show that the video frame interpolation quality of the pyramidal network architecture increases as dual-view correspondence matching is leveraged in more levels, especially for high video resolution, i.e., 2.22 dB PSNR increase and 26.3% LPIPS drop on X-Test. Accordingly, Figure 5 shows the network yields estimates of intermediate frame and flow of lower quality when dropping auxiliary multi-scale collaborative supervision and pyramidal dual-view correspondence matching, implying the proposed approach being critical for more robust correspondence matching by considering neighboring scales to handle large and complex motions.

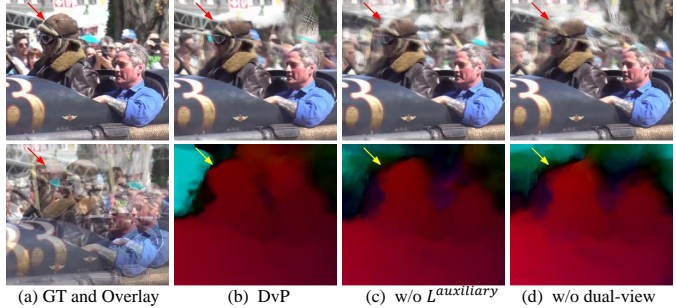

**(a) Overlay**    **(b) EQVI [19]**    **(c) ABME [26]**    **(d) M2M-VFI [11]**    **(e) ST-MFNet [7]**    **(f) SVMV [21]**

**(g) BiFormer [24]**    **(h) EMA-VFI [43]**    **(i) LDMVFI [8]**    **(j) DvP**    **(k) DvP+**    **(l) GT**

**Figure 4: An example of video frame interpolation on SNU-FILM [5] dataset, containing moving wings with feather texture. The proposed DvP and DvP+ models are able to synthesize fine texture details, while previous methods yield artifacts.**

**Table 4: Ablation study on levels of dual-view correspondence matching, deployment of auxiliary multi-scale collaborative supervision loss $L^{auxiliary}$ and construction of pyramidal refinement module. The best numbers are highlighted in bold.**

| Dual-view levels | $L^{auxiliary}$ | Refinement | DAVIS PSNR↑ / SSIM↑ / LPIPS↓ | X-Test (4K) PSNR↑ / SSIM↑ / LPIPS↓ | Params(M) |
|---|---|---|---|---|---|
| × | × | × | 28.03 / 0.8398 / 0.101 | 30.55 / 0.8549 / 0.133 | **5.6** |
| 6 | × | × | 28.10 / 0.8399 / 0.099 | 31.16 / 0.8617 / 0.131 | 5.9 |
| 3-6 | × | × | 28.18 / 0.8401 / 0.095 | 32.25 / 0.8782 / 0.128 | 6.4 |
| 2-6 | × | × | 28.20 / 0.8402 / 0.089 | 32.77 / 0.8801 / 0.098 | 6.6 |
| 2-6 | √ | × | **28.26 / 0.8406 / 0.084** | **33.27 / 0.8972 / 0.068** | 6.6 |
| 2-6 | √ | √ | 28.25 / 0.8392 / 0.082 | 32.98 / 0.8969 / 0.068 | **8.0** |
| 2-6 | √ | + cross-scale guidance | **28.27** / 0.8390 / 0.074 | 33.12 / 0.8978 / 0.066 | 8.0 |
| 2-6 | √ | + attention-based fusion | 28.26 / **0.8393** / **0.072** | **33.35 / 0.8982 / 0.062** | 8.2 |



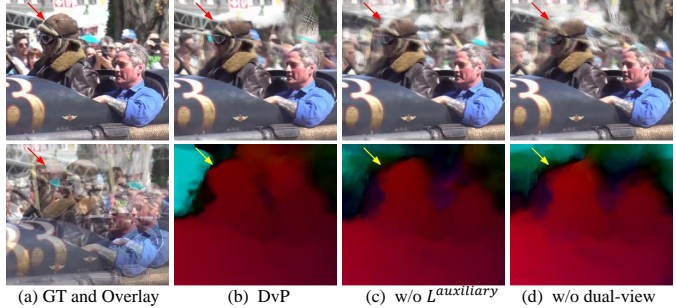

(a) GT and Overlay    (b) DvP    (c) w/o $L^{auxiliary}$    (d) w/o dual-view

**Figure 5: Comparisons of frame (top) and flow (bottom) estimates w/ (b) and w/o auxiliary multi-scale collaborative supervision (c) and dual-view correspondence matching (d).**

**Effectiveness of auxiliary multi-scale collaborative supervision.** We study how the deployment of the proposed auxiliary multi-scale collaborative supervision affects the video frame interpolation performance. To this end, we tested the same pyramidal network architecture without and with the auxiliary multi-scale collaborative supervision being deployed during training, for which results are shown in line 4 and 5 of Table 4 respectively. Without increasing model size, the performance improvements in terms

of PSNR, SSIM and LPIPS are shared by results on DAVIS and X-Test. It shows the efficiency of knowledge exchange to reduce error propagation across scales for varying video resolutions using the proposed auxiliary multi-scale collaborative supervision.

**Effectiveness of pyramidal refinement module.** For evaluation, we add cross-scale flow guidance and attention-based feature fusion, in turn, to a baseline that constructs each scale of a pyramidal module with flow guidance of current scale and convolution-based feature fusion. As shown in line 6 to 8 of Table 4, both cross-scale flow guidance and attention-based feature fusion contribute to improvements of video frame interpolation quality. Such improvements is more obvious on X-Test, implying the eminence of enlarging receptive field for high resolution videos.

## 5 CONCLUSION

In this paper, we propose a pyramidal dual-view correspondence matching algorithm with an auxiliary multi-scale collaborative supervision for video frame interpolation, which can capture complex video dynamics, utilizing the multifaceted knowledge in a pyramid network. The representation capacity of the network can be effectively enhanced with the proposed pyramidal refinement module where we introduce flow-guided cross-scale attention to generate latent representations through efficient feature transform and fusion. Experiments show the superiority of our model in achieving favorable performance, especially perceptual quality.

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
