# OpenReview forum: "Dual-view Pyramid Network for Video Frame Interpolation"
_acmmm.org/ACMMM/2024/Conference — MM2024 Poster_

### Official Review · Reviewer_DjaL · 2024-05-23

**Rating:** 2
**Confidence:** 3

**Summary:**

In this paper, a video frame interpolation method based on dual-view pyramid network is proposed. By introducing dual-view corresponding matching and auxiliary multi-scale collaborative supervision, the performance of video frame interpolation is improved, especially for complex video dynamics.

**Strengths:**

- Introducing pyramidal dual-view correspondence matching to enable robust correspondence between reference frames and the intermediate frame to be synthesized, which captures long-range correlations in the video dynamics.
- Using an auxiliary multi-scale collaborative supervision to calibrate the correspondence matching across scales, which helps reduce error propagation.

**Limitations:**

- The performance is not good enough. On the FILM (Easy) data set, there is still a big gap compared with the best method, which is worth discussing and further improving.
- In introduction part, logical relations are chaotic, and nouns are listed without emphasis, for example, the last paragraph does not clearly describe the key technical details of the proposed method.
- Poor Readability. The equations and sentences in the method are too long to read. At the same time, the symbols lack clear descriptions. The overall logic of the article is poor, and the narrative is not coherent.
- The author provides limited visual results, which are crucial for understanding the performance and perceptual quality of the proposed method.

**Suitability:**

2

---

### Official Review · Reviewer_hj7w · 2024-05-24

**Rating:** 4
**Confidence:** 4

**Summary:**

The proposed method builds a dual-view pyramid network by introducing pyramidal dual-view correspondence matching. The method achieves favorable video frame interpolation performance on several benchmarks with video resolutions ranging from 480p to 4K, while keeping a model size within 10M.

**Strengths:**

Favorable video frame interpolation performance on several benchmarks with video resolutions ranging from 480p to 4K, while keeping a model size within 10M.

**Limitations:**

1. The author claims that video frame interpolation for varying resolution videos remains difficult, where larger motions need to be handled with longer-range correlation analysis while avoiding error propagation in this process, but they do not discuss the framework differences between the proposed method and other extreme video frame compression methods.
2. Teacher-student and multi-scale supervision strategies are commonly used in video frame interpolation, like RIFE[1] and IFRNet[2]. The author seems to have changed the original method to 4 frames input.
3. The experimental performance of the proposed method is compared to SOTA methods, but there is not much comparison between the method with 4 frames of input. It is recommended to display the methods with 2 frames of input and the methods with 4 frames of input separately for a fair comparison. The author can add the comparisons with QVI[3], FLAVR[4], VFIT[5], JNMR[6], MA-CSPA[7].

[1] Huang, Zhewei, Tianyuan Zhang, Wen Heng, Boxin Shi, and Shuchang Zhou. "Real-time intermediate flow estimation for video frame interpolation." In European Conference on Computer Vision, pp. 624-642. Cham: Springer Nature Switzerland, 2022.
[2] Kong, Lingtong, Boyuan Jiang, Donghao Luo, Wenqing Chu, Xiaoming Huang, Ying Tai, Chengjie Wang, and Jie Yang. "Ifrnet: Intermediate feature refines network for efficient frame interpolation." In Proceedings of the IEEE/CVF Conference on Computer Vision and Pattern Recognition, pp. 1969-1978. 2022.
[3] X. Xu, L. Siyao, W. Sun, Q. Yin, and M.-H. Yang, “Quadratic video interpolation,” in Proc. Adv. Neural Inf. Process. Syst., vol. 32, 2019, pp. 1–10.
[4] T. Kalluri, D. Pathak, M. Chandraker, and D. Tran, “FLAVR: Flow-agnostic video representations for fast frame interpolation,” in Proc. IEEE/CVF Winter Conf. Appl. Comput. Vis. (WACV), Jan. 2023, pp. 2070–2081.
[5] Z. Shi, X. Xu, X. Liu, J. Chen, and M.-H. Yang, “Video frame interpolation transformer,” in Proc. IEEE/CVF Conf. Comput. Vis. Pattern Recognit., Jun. 2022, pp. 17482–17491.
[6] M. Liu, C. Xu, C. Yao, C. Lin, and Y. Zhao. “Jnmr: Joint non-linear motion regression for video frame interpolation.” IEEE Transactions on Image Processing, 2023, pp. 5283–5295.
[7] K. Zhou, W. Li, X. Han, and J. Lu, “Exploring motion ambiguity and alignment for high-quality video frame interpolation,” in Proc. IEEE/CVF Conf. Comput. Vis. Pattern Recognit. (CVPR), Jun. 2023, pp. 22169–22179.

**Suitability:**

3

---

### Official Review · Reviewer_oAjE · 2024-05-29

**Rating:** 5
**Confidence:** 2

**Summary:**

The paper proposes a Dual-view Pyramid (DvP) network designed to improve video frame interpolation. It introduces a pyramidal network that utilizes dual-view correspondence matching and a multi-scale collaborative supervision mechanism to enhance the quality of interpolated frames, especially in videos with varying resolutions and complex motions.

**Strengths:**

1) The introduction of dual-view correspondence matching in a pyramidal network is a novel idea and the experiments demonstrate that it tackles the challenge of long-range correlation and error propagation effectively.
2) The approach facilitates the design of a relatively compact model with fewer parameters than the SoTA, which is beneficial for real-world applications where computational resources may be limited.
3) It demonstrates favorable performance on benchmarks with varying video resolutions, indicating the robustness of the method.

**Limitations:**

1) The method section, including the equations and modules in the figures is dense and difficult to follow. It would benefit from clearer explanations and figures to illustrate the proposed method and its main modules.
2) These metrics (PSNR, SSIM, LPIPS), while useful, do not fully capture perceptual quality, especially in complex motion scenarios. The paper could benefit from more diverse evaluation metrics, including user studies or perceptual quality assessments.
3) Test the model on a broader range of datasets and real-world scenarios to better understand its generalization capabilities.

**Suitability:**

3

---

### Meta-Review · Area_Chair_zNQX · 2024-07-06

**Recommendation:** Accept (Poster)
**Confidence:** 4

**Metareview:**

In this submission, the authors propose the Dual-view Pyramid (DvP) network for improving the video frame interpolation with a pyramidal network to enhance the quality of the interpolated frames that utilizes dual-view correspondence matching along with multi-scale collaborative supervision mechanism.

The reviewers agree that there are several strengths in  the submission ( novelty, experimental results, improved performance). They also identify several limitations (some parts of the paper are hard to follow, need for perceptual quality evaluation, results with broader range of datasets, poor readability, marginal improvement in performance, etc). Given the strengths and the weaknesses of the submission, it is recommended to accept the paper assuming tha the readability issues (and any other fixable issues) will be addressed in the camera ready version.